# Investigation of Structural and Tribological Characteristics of TiN Composite Ceramic Coatings with Pb Additives

**Aleksandr Lozovan, Svetlana Savushkina \*, Maksim Lyakhovetsky, Ilya Nikolaev, Sergey Betsofen** 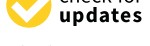 **and Ekaterina Kubatina**

Moscow Aviation Institute, National Research University, 121552 Moscow, Russia; loz-plasma@yandex.ru (A.L.); lyakhovetsky@yandex.ru (M.L.); racer4500@yandex.ru (I.N.); s.betsofen@gmail.com (S.B.); tisaprmp@gmail.com (E.K.)
\* Correspondence: sveta_049@mail.ru; Tel.: +7-(499)-158-4312

**Abstract:** Solid lubricating composite TiN coatings with Pb additives were obtained on steel and titanium substrates in the process of reactive magnetron sputtering of separate cathodes. Columnar, columnar nanostructured and composite nanostructured TiN coatings with different contents (3–13%) of a lubricating component (Pb) were obtained by deposition onto rotating and stationary substrates. It was found that deposition at a rotating substrate and 3% Pb content in the TiN matrix led to a columnar crystallite coating structure. With an increase in its content to 8%, columnar crystallites in the structure become less pronounced, and the coating becomes columnar nanostructured. In nanostructured composite coating with 12% Pb, the soft component is distributed both in the matrix and in the form of inclusions. XRD analysis of the composite nanostructured TiN–Pb coating indicates a textureless state. In this case, the diffraction lines of all present phases (Pb, PbO, TiN) are characterized by a significant broadening, indicating that the size of the subgrains are in range of 10–20 nm. Tribological tests of the coatings were carried out at room temperature and under conditions of stepwise heating. The nanostructured composite coating showed the best tribological characteristics due to a high Pb content, a relatively high microhardness (817 HV) and a textureless state with a low grain size. This coating had a low friction coefficient (~0.1) over 50,000 test cycles, both at room temperature and under conditions of stepwise heating up to 100 °C and 200 °C.

**Keywords:** solid lubricant coating; titanium nitride; lead; nanostructured coating; composite coating; tribological testing; friction coefficient; stepwise heating



## 1. Introduction

An increase in the loads of thermal machines and the desire to reduce the weight of products lead to a gradual replacement of the liquid lubricants system in favor of alternative solutions. Solid lubricant coatings (SLC) can resist wear for a long time at elevated temperatures and high contact pressures [1–3]. Such coatings should have high wear resistance to ensure a long service life and a low friction coefficient. Various combinations of oxide, nitride, and carbide ceramics, wear-resistant metals, and intermetallic compounds are promising materials as a matrix for SLC [4–7].

Solid lubricants currently used at high temperatures can be divided into three categories [8,9]: (1) soft metals (Ag, Cu, Au, Pb, In and etc.); (2) fluorides (e.g., $CaF_2$, $BaF_2$ and $CeF_3$); and (3) metal oxides (e.g., $V_2O_5$, $Ag_2Mo_2O_7$). All three types of materials plastically deform and/or form low shear strength surfaces at elevated temperatures. They do not lubricate at relatively low temperatures, so they have been combined with low temperature lubricants to create "chameleon" coatings that adapt their surface during a temperature cycle from 25 °C to 1000 °C to reduce friction in this temperature range [1,2,10].

At present, various magnetron sputtering types are increasingly used for tribological coatings deposition in industry. The method makes it possible to vary the microstructure

of coatings, to change the grain size, phase composition and crystallographic orientation that creates a control mechanism in a wide range of coating properties. The most effective modern approaches of SLC properties increase are: (1) deposition of multicomponent coatings, when, along with the main metal component (for example, Ti, Zr), such elements as Al, Cr, Nb, Y, Si are introduced into the coating; (2) multilayer coatings containing alternating layers with the thickness from several nanometers to micrometers; (3) combined deposition methods accompanied with ion nitriding or ion implantation. In recent years, multicomponent coatings have attracted increased interest [11–18]. For example, in [11] it was shown that titanium nitride-based coatings doped with Al and Zr had higher heat resistance due to the formation of stable, dense oxides on the surface, which increased the performance of coated products at elevated temperatures. For a number of systems, it was found that three-component coatings had higher hardness and wear resistance compared to binary coatings based on these elements [12]. The addition of metalloids atoms with the formation, in particular carbonitride and oxynitride coatings, affect their properties in a more complex way. Generally, carbonitride coatings have a higher hardness and oxynitride coatings have a lower hardness compared to nitrides. Some elements (e.g., Y and Si) added in amounts up to 10% lead to amorphization of the coating structure [7,13–16]. For example, in works [7,12], the addition of Al and Si in TiN coatings was accompanied by a change in the crystallographic texture. The pronounced (111) texture of TiN coatings passed into a state close to textureless. In textureless TiN and ZrN coatings, the hardness was two times higher than in coatings with a pronounced (111) texture [19].

The authors of work [2] suppose that oxides are potentially the best choice for SLC under extreme conditions at high temperature since oxides are often structurally and chemically thermodynamically stable although there are exceptions. Solid solutions such as $(Al,Cr)_2O_3$ form a corundum phase in relatively low temperature PVD processes and may be promising for low friction and wear coatings, but they are metastable at moderate temperatures [20]. Recently, researchers have focused on understanding the tribological properties of various Magneli phases, as $WO_x$, $VO_x$, $MoO_x$ and $TiO_x$ [21].

In work [2], the authors added a silver to a binary oxide to create ternary oxides. Silver is a soft metal and self-lubricating at relatively high temperatures (T > 300 °C). Ag–O bonds are relatively weak. They enhance the shearing of crystal planes and lower the melting point of initial metal double oxides when incorporated into the ternary oxide crystal structure. Besides, elemental Ag does not easily oxidize and will improve the tribological properties of the surrounding oxide phases by increasing the toughness of the film. Ag adds effects to a solid nitride matrix. Its influence to tribological properties at different operating temperatures and the transport activity of silver depending on the deposition temperature conditions were studied by authors of works [22–26].

The concept of tribological oxidation with the formation of ternary oxides was first described in [27]. In this research, composite $PbO$–$MoS_2$ coatings, which formed a $PbMoO_4$ lubricating layer at elevated temperatures, were obtained. In a subsequent study focusing on the lead molybdate phase [28], the authors found that, although this material was abrasive at low temperatures, it showed reduced friction coefficients at high temperatures (~0.3–0.4 at 700 °C) [18]. It has been found that the presence of a soft metal with low oxygen bond strength in ternary oxides was effective decision for high temperature solid lubricants. The low bond strength results show that low melting point materials are soft and easy to shear and can degrade into a binary oxide and a soft metal to improve lubricity. Recently, such soft metals as Pb, Cu and In have attracted interest.

Thus, studies of the Ti-N-Cu system with the aim of developing both multilayer and composite wear-resistant coatings were carried out using various options of vacuum-arc [29] and magnetron [30,31] deposition and their hybrid application [32,33].

Copper is a soft metal with excellent thermal conductivity contributed to efficient dissipation of friction heat and therefore lower temperatures in the contact area. The main lubrication mechanism is their increased ductility and low shear strength at high temperatures. Thus, soft metals can plastically deform during sliding and conform to both

interacting surfaces, reducing friction and wear. Increased softening at high temperatures can lead to the pulling of soft metals from the interface, which limits their lubricating effect [34].

Lead has excellent tribological properties. In works [35,36], the magnetron sputtered with ion assistance TiN based solid lubricant coatings with the Pb addition showed the advantage of composite structure over a multilayer structure with alternating layers of a solid matrix and lubricant component. The development of these works has shown that an increase in the lubricant component in the matrix leads, on the one hand, to a decrease in the friction coefficient and, on the other hand, reduces the wear resistance of the coating.

In [37], the tribological behavior of PVD TiN coatings with the addition of indium was studied. Tribological studies showed an improvement of wear resistance over unmodified TiN thin films up to 450 °C. Observed deterioration in coating behavior at higher temperatures was attributed to indium oxidation.

In this work, composite TiN based SLC with different structure and Pb content have been formed by Ti and Pb cathodes co-sputtering of two separate magnetrons, and their tribological properties have been studied at room and elevated temperatures.

## 2. Materials and Methods

Solid lubricant nanostructured TiN coatings with Pb additives were formed on steel AISI 304 and titanium VT1 (99.0% Ti) samples in the process of reactive magnetron sputtering of separate cathodes (99.2% titanium, 99.5% pure lead).

Planar magnetrons with the size of both targets, 0.273 m × 0.112 m × 0.01 m, were placed vertically in the chamber at different distances d from the target to the substrate during the deposition of TiN–Pb coatings (Figure 1). The ion source was placed vertically opposite the substrate at a distance $d_1$ from it (Table 1). The screens made of steel were located next to them to reduce the possible transfer of sputtered atoms from one magnetron to another. Ar and N gases were fed into the vacuum chamber through an ion source.

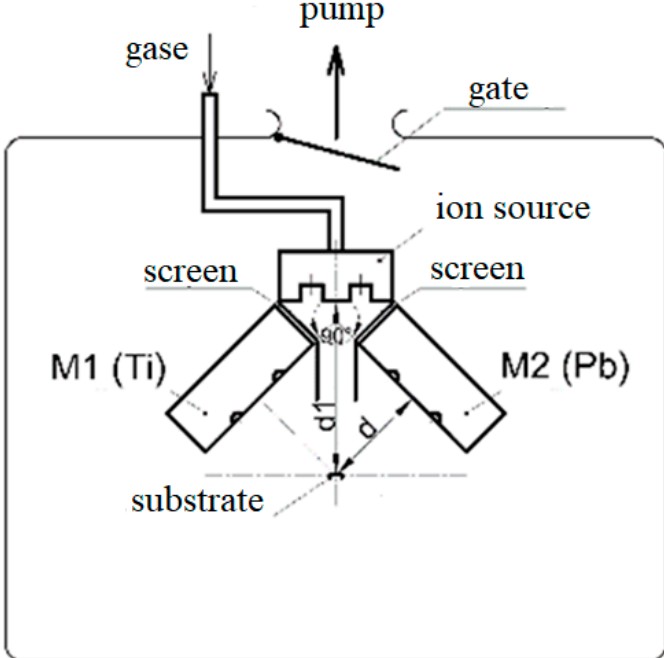

**Figure 1.** Scheme of TiN–Pb coatings deposition.

**Table 1.** Parameters of the coating deposition process. $Q_{Ar}$, $Q_{N2}$—gas flow rates; $I_{Ti}$, $I_{Pb}$—discharge currents; t—spraying duration; F—power supply frequency; d—target-substrate distance; $d_1$—substrate–ion source distance; n—substrate rotation speed.

| No. | Substrate | $P_{Ar+N2,}$ Pa | $Q_{Ar,}$ cm$^3$/min | $Q_{N2,}$ cm$^3$/min | $I_{Ti}$, A | $I_{Pb}$, A | t, min | F, kHz | d, mm | $d_1,$ mm | n, rpm |
|-----|-----------|------|------|------|-----|-----|-----|----|-----|-----|---|
| 1 | steel | 0.31 | 8.65 | 2.45 | 3.5 | 0.1 | 720 | - | 100 | 160 | 2 |
| 2 | steel | 0.25 | 8.54 | 4.1 | 3.5 | 0.1 | 350 | - | 100 | 160 | 2 |
| 3 | steel | 0.24 | 6.52 | 5.14 | 3.5 | 0.1 | 350 | 25 | 220 | 250 | - |
| 4 | steel, Ti | 0.24 | 6.49 | 5.18 | 3.5 | 0.1 | 350 | 40 | 220 | 250 | - |

The samples were cleaned in an ultrasonic bath in gasoline for 10–15 min before spraying. Then, they were placed into the chamber, and it was pumped out to a pressure of $5.3 \times 10^{-4}$ Pa. The samples were cleaned using an ion source for 20 min at $P_{Ar} = 0.13$ Pa at a flow rate of $Q_{Ar} = 6.49$ cm$^3$/min. Then, the ion source was turned off. Ti and TiN transition layers were deposited for 10 min. Then, the main TiN + Pb coating layer was deposited. The main parameters of the coating deposition process are given in Table 1. Ti and Pb sputtering was carried out in the constant current mode with current stabilization when coatings 1 and 2 were deposited. For coatings 3 and 4, Ti sputtering was carried out in the constant current mode with current stabilization, and Pb sputtering was carried out in the medium frequency mode at 25 and 40 kHz and a duty cycle T = 80%. The total deposition time was 350 min for coatings 2–4 and 720 min for coating 1. During the deposition of samples 1 and 2, the substrate was rotated clockwise at a speed of 2 rpm.

The morphology and composition of the coatings were studied using a Quanta 600 scanning electron microscope (SEM, FEI Company, Eindhoven, The Netherlands) equipped with a TRIDENT XM4 energy dispersive X-ray microanalysis system. The thickness of the coatings was measured in a cross-section of the samples using SEM. The roughness was examined using an Olympus LEXT OLS 5000 confocal microscope (Olympus, Tokyo, Japan). 2D profiles were used to obtain the roughness parameters. X-ray phase analysis was performed by DRON-7 X-ray diffractometer (NPP "Burevestnik", Saint-Petersburg, Russia) in filtered CuK$\alpha$ radiation with a wavelength $\lambda = 1.54178$ Å. The microhardness of the coatings was evaluated on a Buehler Micromet 5101 instrument (Buehler, Lake Bluff, IL, USA) by indenting a Vickers pyramid with a load of 50 g.

Ball-on-disk tribological tests were carried out in the reciprocating wear mode with a displacement of 15 μm, a normal contact load of 1 N, a displacement frequency of 20 Hz, a cycles number of 50,000 (total sliding distance—1.5 m), an ambient temperature of 23 °C and a humidity of 37 ± 5%. A sphere with a diameter of 12.6 mm made of 100Cr6 steel (62–65 HRC) was used as a counter body. The influence of heating on the tribological properties of the coatings was studied in the stepwise heating mode up to 100 °C and 200 °C with holding at each temperature for 2 h. The coated samples were tested to determine the friction coefficient after each stage of heating.

Volumetric wear was analyzed using a confocal laser microscope LEXT OLS5000. The microscope software evaluates volumetric wear by integrating wear worn space points relative to the original surface. The transferred material of the counter body was estimated as the build-up volume in the wear patches. The build-up height exceeded the original level of the coating surface. In the case of its formation, the build-up volume corresponded to the volumetric wear of the counter body.

Also, tribological tests of the samples were carried out with a change in the loading parameters: displacement D = 5–60 μm; normal contact load F = 1–13 N; displacement frequency f = 20 Hz; the number of cycles n = 10,000. The friction coefficient and the mechanism of bodies interaction were analyzed in the testing process.

## 3. Results

### 3.1. Surface Structure and Elemental Composition

The coatings surface morphology on a steel substrate is characterized by globules with the size of ~1–5 μm (Figure 2). Coating 1 has the largest surface globules (Figure 2a), which may be due to the longer deposition time. The smallest size of surface globules was obtained for coating 3 (Figure 2c), while their highest packing density was obtained for coating 4 (Figure 2d). An enlarged image of the coating 4 globule structure is shown in Figure 2e. Differently oriented crystallites less than 0.1 μm in size are seen in the structure of the globule. Globules and cavities on the coatings surface could be formed in the presence of high compressive stresses. Thermal stresses arising as a result of temperature changes at the deposition completion can promote diffusion processes. In such a way, mass transfer of the deposited material can occur [38].

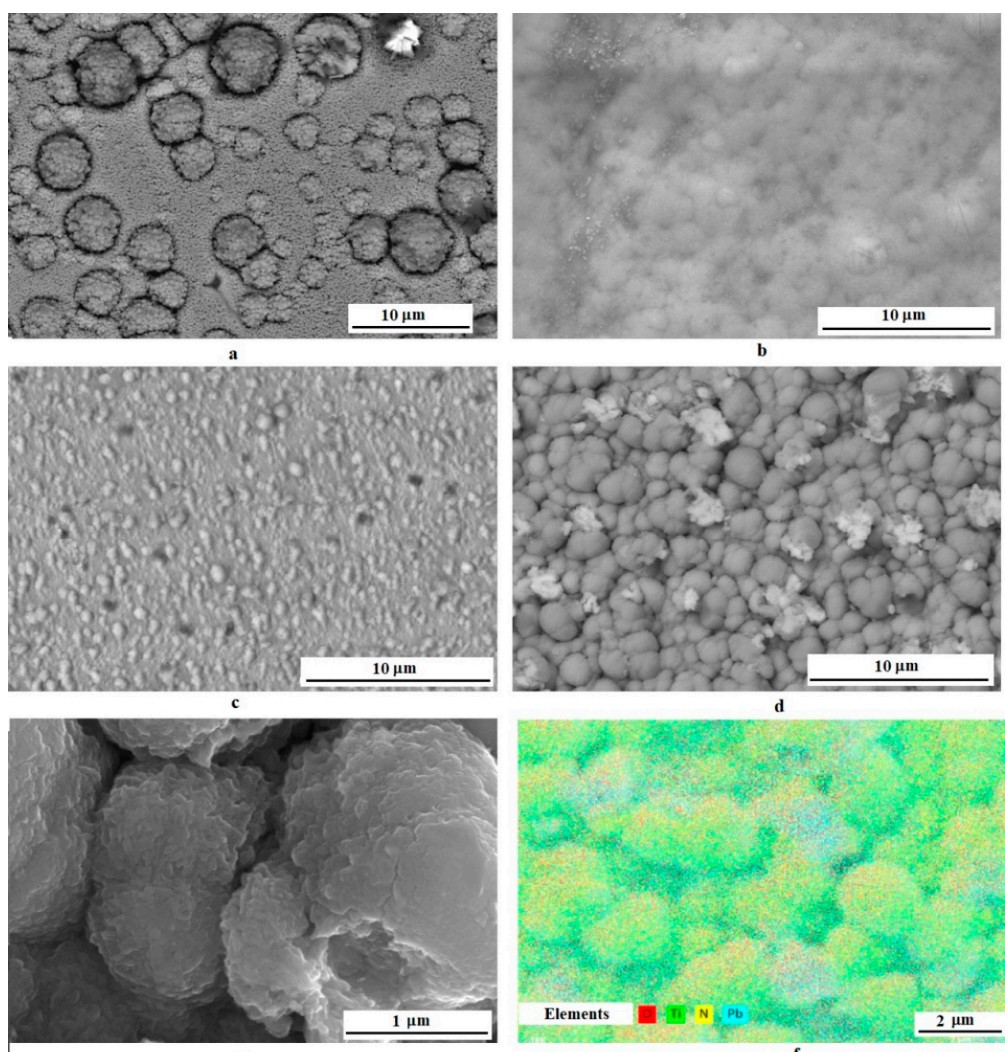

**Figure 2.** SEM images in backscattered electrons of the surface for coatings 1 (**a**), 2 (**b**), 3 (**c**), 4 (**d**), an enlarged image in secondary electrons of coating 4 globules (**e**) and a summary map of the elements distribution over the coating 4 surface (**f**).

X-ray microanalysis showed a uniform distribution of Pb over the surface for coatings 1–3 and the presence of areas with an increased content of Pb and O for coating 4 (Figure 2f). The thicknesses of coatings 1–4 are ~5.8, 3.8, 2.3 and 2.0 μm, according to the results of SEM studies on cross-sections. The average lead content is ~3, 8, 13 and 12 at.% for coatings 1–4, respectively. With an increase in the $Q_{Ar}/Q_{N2}$ ratio, the TiN content in the coating

increases, and the Pb content decreases. The oxygen content in the surface layer of the coatings increases with an increase in the Pb content from 12 at.% (coating 1) to 40 at.% (coatings 3 and 4), which is associated with its oxidation.

### 3.2. Surface Roughness

In Figure 3, the coatings surface height maps obtained using a confocal microscope are shown. For coatings 1–3, the surface topography changes from the predominance of protrusions to a greater content of depressions with an increase in the lead content from 3 to 13%. A decrease in the roughness parameters $R_a$ and $R_q$ to 0.126 and 0.464 μm for coating 3 (Table 2) shows that the surface is getting smoother. $R_a$ parameter and, more significantly, Rq parameter decreased with an increase in the lead content in the coatings, which may be due to the smoothing of the surface as a result of the lower thermal stresses action after the deposition process completion. The parameters $R_a$ and $R_q$ are higher for coating 4 with an uneven distribution of lead in the surface layer than for coatings 1–3. The roughness parameters $R_{sk}$ and $R_{ku}$ analysis allows the prediction of the tribological behavior of the coatings. $R_{sk}$ parameter shows the asymmetry of the height distribution, and $R_{ku}$ estimates the flatness or sharpness of the surface topography. Since the values of $R_{sk} > 0$ for all coatings, the protrusions dominate in the structure of all coatings. The smallest $R_{sk}$ was obtained for coating 4, which indicates a significant proportion of depressions on the surface. The parameter $R_{ku}$ is high for 1 coating due to individual sharp protrusions. $R_{ku}$ shows a significant smoothing of the surface due to the reduction of the "sharpness" of the protrusions for coatings 2–4. Coating 1 is characterized by the highest values of $R_{sk}$ and $R_{ku}$, which shows the predominance of higher and "sharper" protrusions on the surface. It was found in [39] that the following values of roughness parameters are satisfied to maintain an oxide coating low friction coefficient for a long time: $R_a = 0.4$ μm, $R_{sk} = -1.8$ and relatively high $R_{ku} = 10$. Thus, from the point of view of the obtained roughness parameters, coating 4 can have the lowest friction coefficient due to both a relatively low value of $R_a$ and the lowest $R_{sk}$.

### 3.3. Cross-Section Coatings Structure

The coatings have a two-layer structure consisting of a Ti + TiN transition layer (1) up to 0.1 μm thick and the main coating TiN + Pb layer (2) (Figure 4). The inhomogeneity of the coating thickness is due to the globular structure of the surface. The globules height reaches 0.3 μm. Coating 1 has a columnar structure consisting of intergrown crystallites located perpendicular to the surface (Figure 4a). Such structure is typical for deposited TiN. It is characterized by the (111) texture component predominance [19]. In growth process, the deposited atoms diffuse over the surface until they enter the low-energy lattice sites and are included in the growing coating. The deposited atoms can change their positions in the crystal lattice due to diffusion and recrystallization processes [38]. The thickness of columnar crystallites (0.05–0.25 μm) increases with approaching to the surface of the coating. Their magnified image is shown in Figure 4a. The elements distribution map of coating 1 cross-section showed a uniform distribution of Pb in depth mainly along columnar crystallites, which may be due to the filling of intercrystallite spaces by it (Figure 4b). Coating 2 is columnar nanostructured. Columnar crystallites become less pronounced. Their thickness and length decrease, but a noticeable texture is retained, as in coating 1 (Figure 4a). The structure of coating 3 corresponds to the textureless state (Figure 4d). Columnar crystallites disappear, the structure becomes denser, nanocrystallites appear, which suggests the coating amorphous-crystalline state. Thus, for coatings 1–3, the structure with an increase in lead content from 3 to 13% changes from columnar to dense textureless. Coating 4 is composite nanostructured. On the one hand, its state is also close to textureless, but unlike coating 3, it consists of differently oriented nanocrystallites (Figure 4f). Lead is present both in the coating matrix and in the form of islands predominantly distributed in the surface layer of coating with a thickness of ~0.6 μm (Figure 4f). At the same time, its average content is less than that of coating 3. This suggests a smaller amount of Pb in the

matrix compared to coating 3, which may contribute to the nanostructured state. A special metastable state for the process of coating formation can lead to phase migration of the soft and fusible component and its higher concentration in the surface coating layer [40].

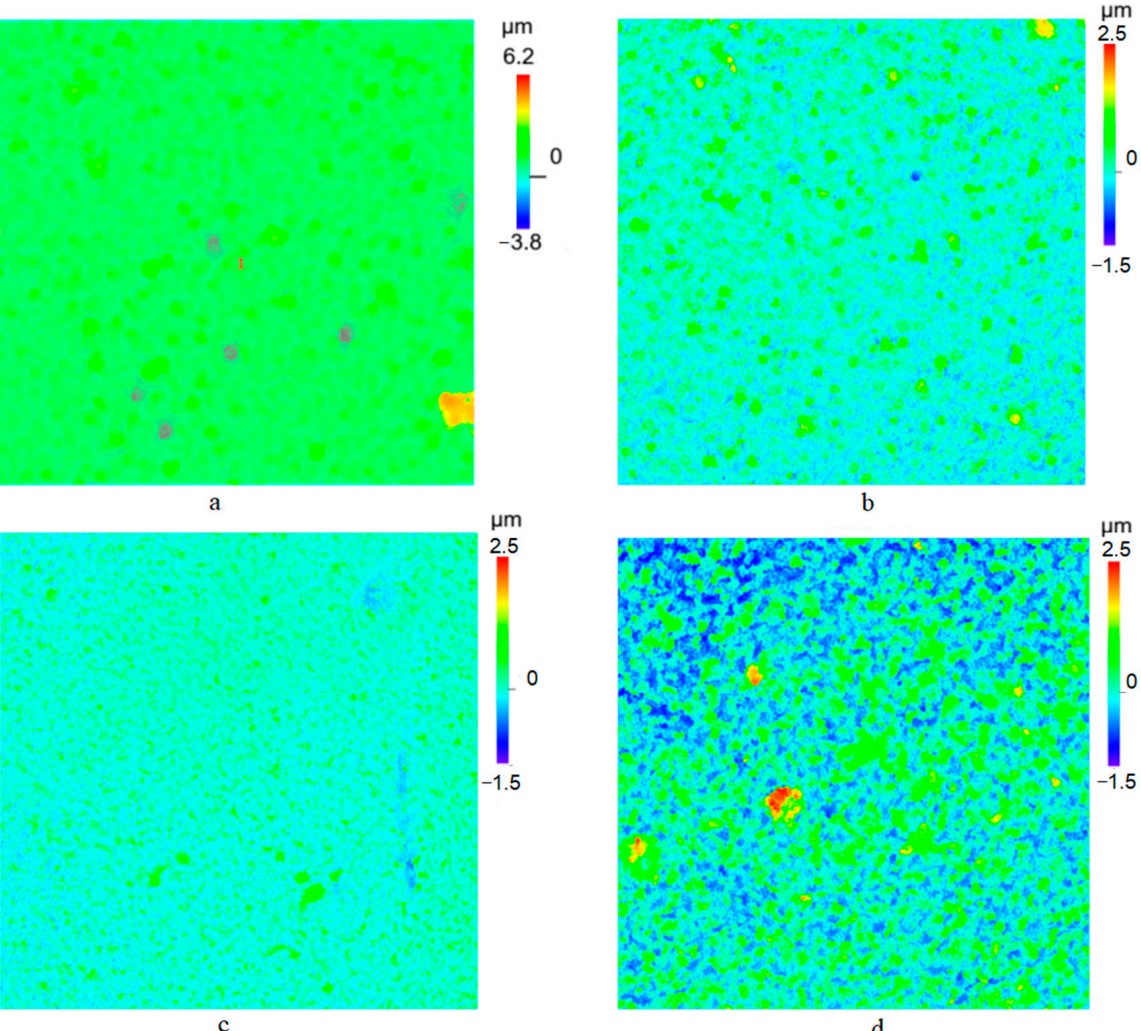

**Figure 3.** Surface height maps of coatings 1 (**a**), 2 (**b**), 3 (**c**), 4 (**d**) on a steel substrate.

**Table 2.** Roughness parameters of coatings 1–4.

| No. | Pb, at.% | $R_a$, µm | $R_q$, µm | $R_{sk}$ | $R_{ku}$ |
|---|---|---|---|---|---|
| 1 | 3 | 0.277 | 0.486 | 4.265 | 44.251 |
| 2 | 8 | 0.187 | 0.258 | 1.657 | 8.246 |
| 3 | 13 | 0.126 | 0.170 | 1.382 | 6.427 |
| 4 | 12 | 0.366 | 0.464 | 0.739 | 4.014 |

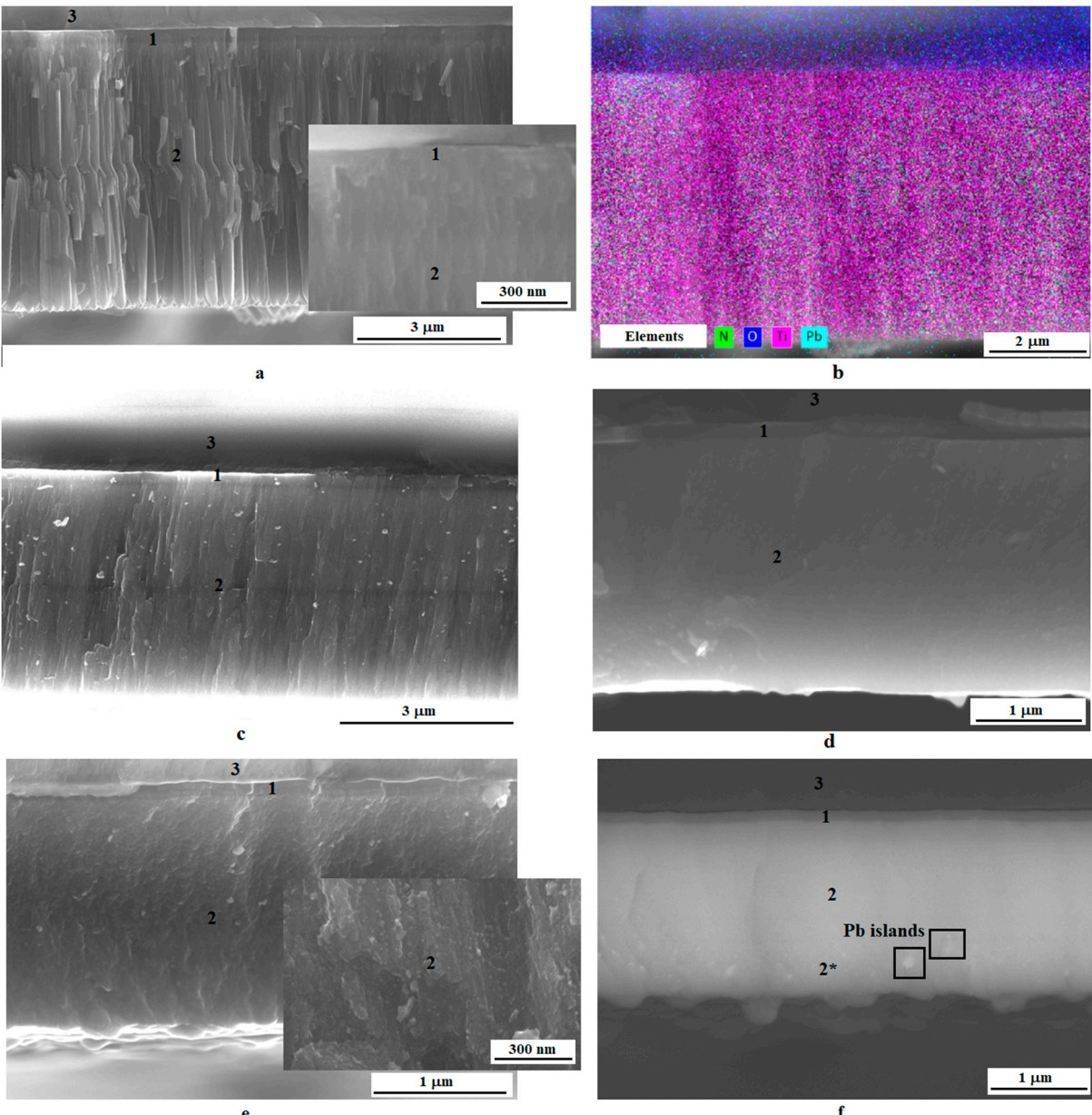

**Figure 4.** Cross-section structure SEM images in secondary electrons of the coatings 1 (**a**), 2 (**c**), 3 (**d**), 4 (**e**), a summary map of the elements distribution for coating 1 (**b**), SEM image of coating 4 in backscattered electrons (**f**). 1—Ti + TiN transition layer, 2—main TiN + Pb layer, 2*—surface layer with island-like Pb-containing inclusions, 3—substrate.

### 3.4. Phase Composition

Coatings 1 and 2 have a pronounced columnar structure. Usually such structure has pronounced (111) crystallographic texture. The XRD pattern of TiN–Pb coating 4 (Figure 5) indicates a textureless state. The diffraction lines of all present phases (Pb, PbO, TiN) are characterized by a significant broadening, indicating that the size of the subgrains are in range ~10–20 nm. It can be assumed that the absence of a columnar structure is associated with its discontinuous growth provided by Pb because it does not dissolve in the TiN matrix and has a weak tendency to nitridation. Formation of Pb and PbO phases nuclei on the surface of TiN crystallites promotes their growth interruption and their nanometer size retention. In addition, particles of Pb and PbO phases stimulate the formation of TiN nuclei of random orientations, which prevents the formation of a pronounced texture.

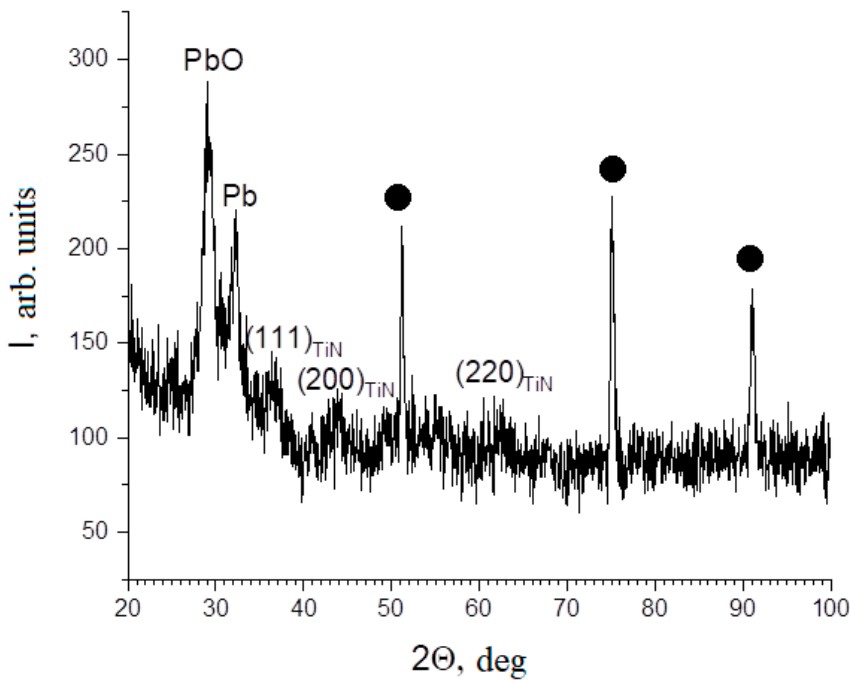

**Figure 5.** XRD pattern of TiN–Pb coating 4 on a steel substrate (●).

It was shown in [41] that an increase in the (111) texture component of TiN coating is accompanied to decrease in wear resistance. The textureless structure of the TiN coating may contribute to an increase in microhardness and wear resistance. Discontinuous structure and grain reduction provide strengthening in accordance with the Hall-Petch law.

*3.5. Microhardness*

The highest Vickers microhardness (919 $HV_{50}$) was obtained for columnar coating 1 with the greatest thickness (Table 3). The transition of the coating structure to a columnar nanostructured and textureless state, as well as a decrease in thickness, led to a decrease in microhardness. This is also associated with an increase in the content of the soft Pb component in the structure. For coating 3, the microhardness values become similar with those for the substrate—steel. An increase in Pb content can lead to the formation in the coating structure of such compounds as lead oxides, lead oxynitride, $Ti_3PbO_7$, $TiPbO_3$, etc. [42]. The hardness of these compounds is significantly inferior to TiN, which contributes to a general decrease in hardness, despite the compaction of the coating structure and a decrease in the size of crystallites. Despite its more than 2 times smaller thickness, the microhardness of the composite nanostructured coating 4 is 817 HV, which is not much inferior to the columnar coating. This must be due to the composite structure of the coating. In coating 4, surface layer lead is mainly present in the form of island inclusions, which makes it possible to maintain a solid matrix. The content of large Pb inclusions increases when approaching the surface. In addition, grains' size reduction provides strengthening in accordance with the Hall-Petch law.

*3.6. Tribological Tests at Room Temperature*

In tribological tests, the lowest friction coefficient (~0.1) was shown by nanostructured composite coating 4 (Figure 6). Its friction coefficient practically does not change during 50,000 test cycles. Textureless coating 3 also had low friction coefficient at the initial stage of testing. Already after 2000 cycles, it increases sharply, which is associated with the onset of the coating destruction. During subsequent cycles, the friction coefficient increases from 0.3 to 0.35. For columnar coating 1 and columnar nanostructured 2, the friction coefficient is ~0.27 during all 50,000 test cycles. Intensive destruction was accompanied by large oscillations in the friction coefficient. Small rises and decreases in the friction coefficient

during the experiment are associated with the accumulation of wear products in the contact zone and their removal outside the contact patch.

**Table 3.** Structure, Pb content and microhardness of the coatings.

| No. | Thickness, μm | Pb, at.% | Structure | Microhardness, $HV_{50}$ |
|---|---|---|---|---|
| 1 | 5.8 | 3 | Columned | 919 |
| 2 | 3.8 | 8 | Columned nanostructured | 570 |
| 3 | 2.3 | 13 | Textureless | 283 |
| 4 | 2.0 | 12 | Nanostructured composite | 817 |

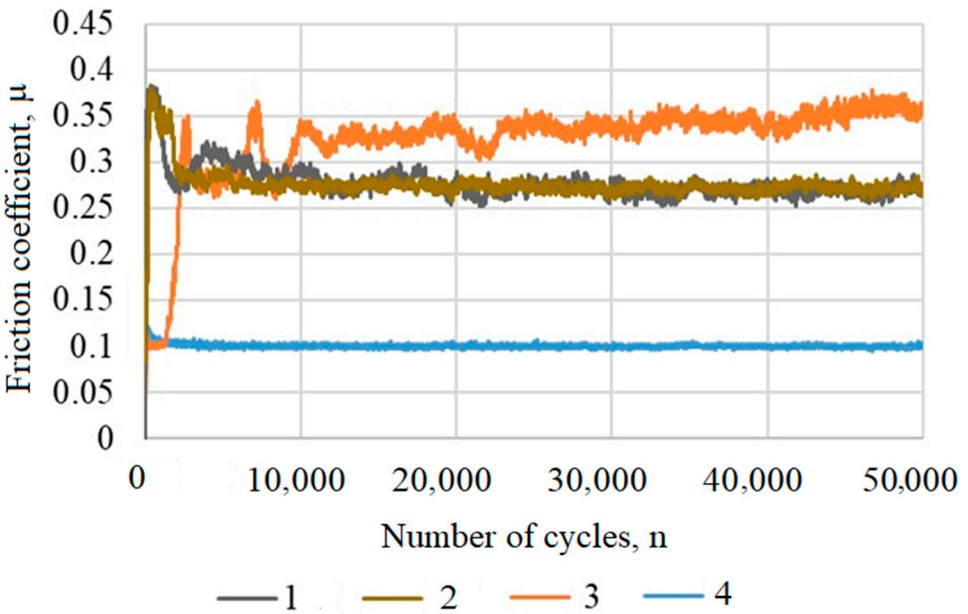

**Figure 6.** Comparison of friction coefficients of coatings 1–4 on a steel substrate when moving at 15 μm.

Figure 7 shows a comparison of the volumetric wear of the coatings and the material transfer from the counter body. An increase in the height of wear spots relative to the main coating and an increase in the friction coefficient indicated significant transfer of material from the counter body (steel) for coatings 1 and 2. Such a significant mass transfer should be associated with the rapid destruction of the coatings and the interaction of the substrate with the counter body, as can be seen in the graphs of the friction coefficient. "Sticking" was not observed for coatings 3 and 4. However, the volumetric wear of coating 3 is much higher.

### 3.7. Tribological Tests under Stepwise Heating Conditions

The effect of temperature on the tribological properties of the coatings was carried out with stepwise heating to 100 °C and 200 °C and a holding time of 2 h in a muffle furnace at each temperature. After heating tribological tests were carried out (Figure 8). Composite nanostructured coating 4 demonstrated the stability of tribological characteristics. The friction coefficient remained an approximate value of 0.1 during 50,000 test cycles, as after heating to 100 °C. After heating to 200 °C, it slightly increased up to 0.12. After heating to 100 °C, the friction coefficient of coating 3 increased already to 0.4 at the beginning of the tests (Figure 8a). After subsequent heating, it decreased to 0.3. Increased softening at high

temperatures can cause the soft component to be pulled away from the interface, which limits its lubricating effect.

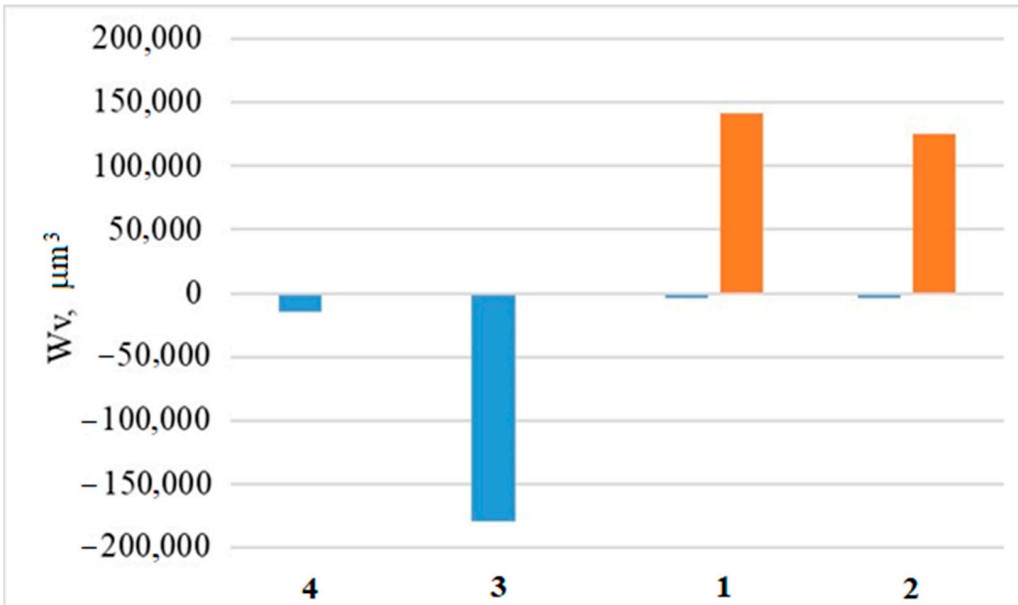

**Figure 7.** Volumetric wear of the coatings (blue columns) and material transfer from the counter body (orange columns).

Figure 9 shows images of wear patches. After heating to 200 °C, the coatings were subjected by rather strong oxidation that affected their tribological properties. Minor improvement in coating 3 tribological characteristics at the beginning of the tests could be due to the formation of oxides and oxynitrides of the system components.

*3.8. XRD Analysis after Stepwise Heating*

Some change in the properties of the coatings may be associated with the oxidation of the TiN–Pb system components, as well as the substrate. After heating reflections of Pb with FCC lattice and reflections of TiN are retained in addition to reflections from the substrate in XRD pattern of coating 4 (Figure 10). Reflection angles $2\Theta{\sim}36°$ correspond to reflections (200) Pb and (111) TiN, and reflection angles $2\Theta{\sim}62°$ correspond to reflections (311) Pb and (220) TiN, which cannot be uniquely identified as belonging to Pb or TiN.

The ratio of Pb reflection intensity ($2\Theta{\sim}36°$) to PbO ($2\Theta{\sim}28°$) increases compared to the diffraction pattern before heating (Figure 5). The diffraction lines of Pb, PbO and TiN are strongly broadened, which indicates the nanostructured state retention. A decrease in the signal-to-background ratio for Pb reflections compared to the XRD pattern before heating may indicate the formation of other oxide compounds with lead. In addition, oxygen atoms can also enter the TiN matrix, forming titanium oxynitride, which retains the original crystal structure (NaCl type). In this case, oxygen atoms replace some of the nitrogen atoms without forming $TiO_2$, and the signal-to-background ratio in the region of these reflections may decrease [43].

After heating, the intensity ratio of $I_{Pb}/I_{PbO}$ reflections is much higher for coating 3 than for coating 4, which indicates less oxidation. Most likely, this is due to a more uniform distribution of Pb in the matrix and the absence of islands with its increased content in the surface layer of the coating.

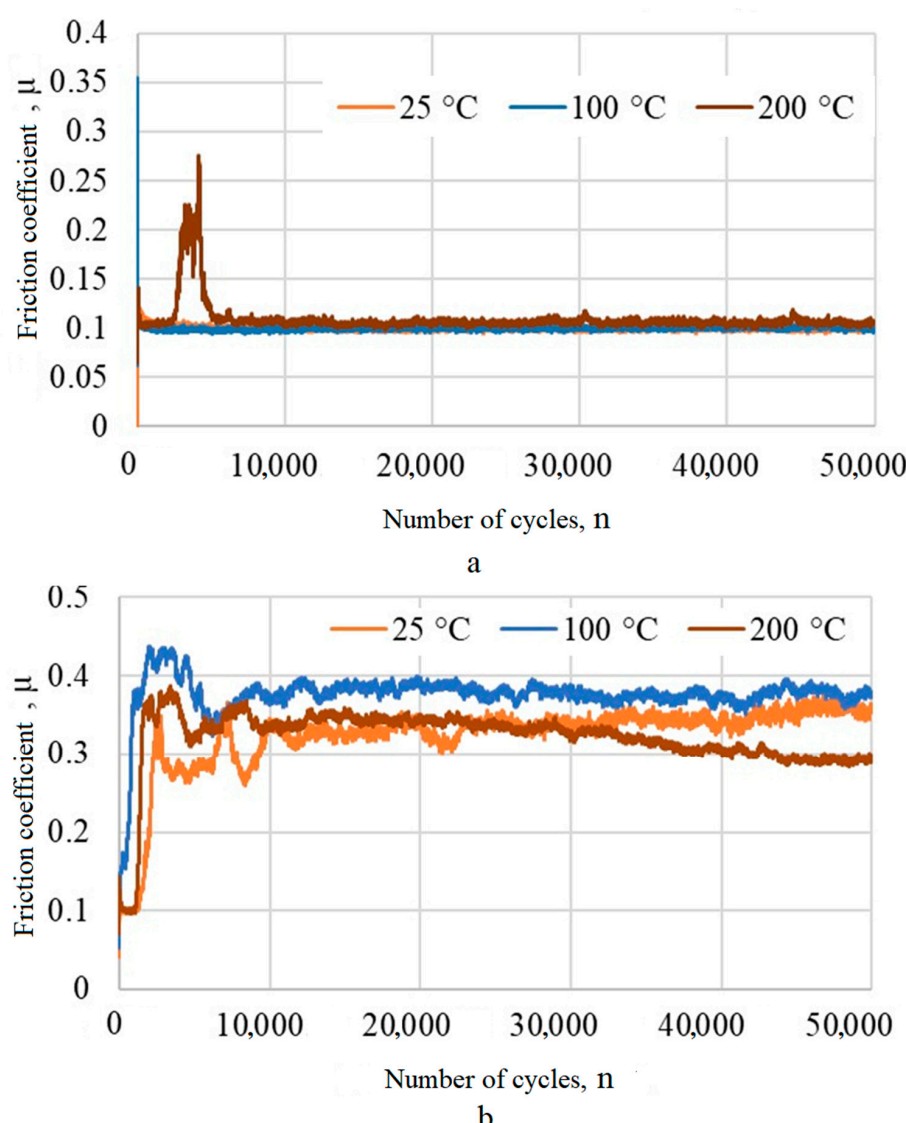

**Figure 8.** Friction coefficients of coatings 4 (**a**) and 3 (**b**) on a steel substrate during stepwise heating.

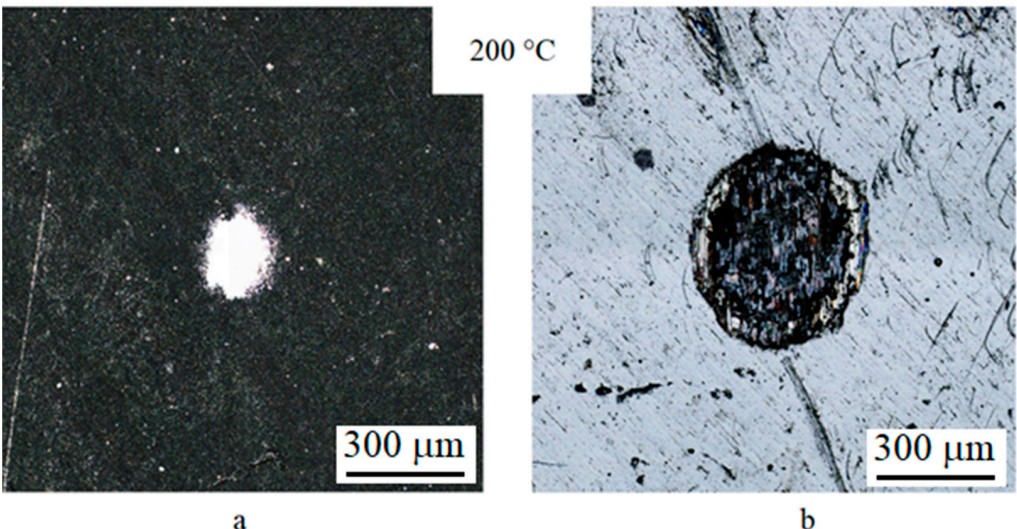

**Figure 9.** Images of wear patches after heating to 200 °C for coatings 4 (**a**) and 3 (**b**).

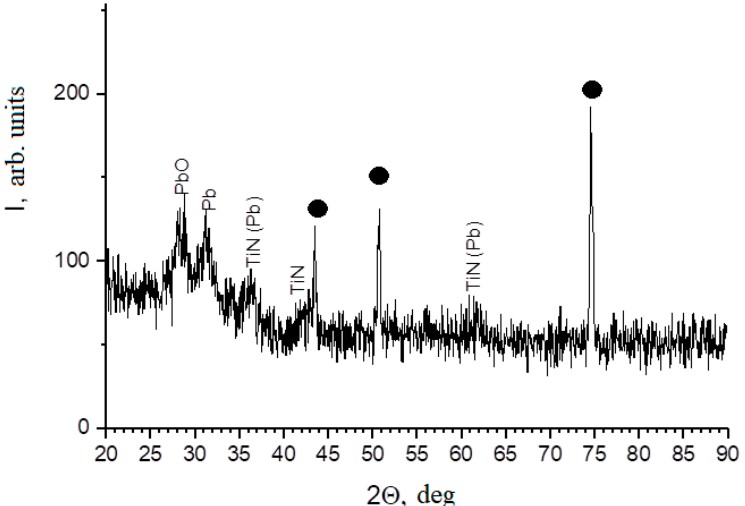

**Figure 10.** XRD pattern of coating 4 after stepwise heating up 200 °C. ●—steel substrate.

*3.9. Tribological Study of a Nanostructured Composite Coating at Various Loading Parameters*

　　Coating 4 was studied in friction modes that differ in mechanisms of surface destruction, when fatigue processes, abrasive damage by wear products or adhesive wear can predominate [44,45]. The mechanism of friction during fretting was analyzed using an energy approach [46]. It consists in dissipation energy determining in the contact by measuring the instantaneous values of the friction force and displacement with a frequency at least 20 times higher than the frequency of bodies movement. The fretting hysteresis loops analysis makes it possible to estimate the modes of friction bodies interaction directly in the course of the experiment by calculating the slip index. Tribological tests were carried out in a wide range of loading parameters to obtain a fretting map. The resulting fretting map based on the hysteresis loops analysis is shown in Figure 11.

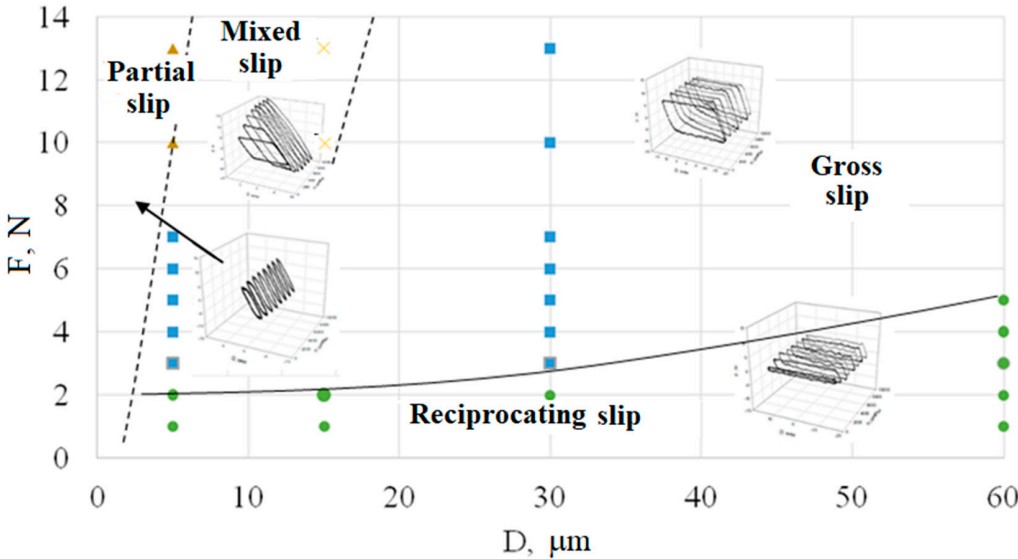

**Figure 11.** Fretting map of nanostructured composite coating 4 on titanium.

　　The process of abrasive and adhesive interaction of the bodies prevails in gross slip mode during fretting. The gross slip mode is indicated by blue squares in Figure 11. Fast fatigue failure dominates in mixed slip mode (yellow crosses). Fatigue failure with low cracking rate is observed in partial slip mode (brown triangles). The friction processes without the specific features of the friction bodies interaction in fretting corresponds to the

reciprocating slip mode (green circles). The wear patch obtained at D = 5 µm and F = 13 N is an uneven damage with dimensions of ~100 µm along the displacement axis and ~250 µm across (Figure 12a). The areas with a high lead content are visible on the coating surface in the form of light spots. Cracks up to 15 µm long are observed on the surface of the coating. They are located perpendicular to the direction of the bodies' friction (Figure 12b). The coating mechanism destruction corresponds to the process prevailing for the partial slip mode–fatigue failure. It occurs as a result of the sign-alternating friction force action in a contact in combination with the transition between slip zones and elastic bodies interaction. At loading parameters (D = 15 µm, F = 10 N), the process of fatigue failure is significantly getting worse. A network of cracks is formed over the entire area of the contact patch on the coating surface, which corresponds to a mixed slip mode (Figure 12c). The process of further destruction can lead to flaking of large agglomerates of the coating during friction and interaction of the counter body with the substrate. Also, under certain conditions the cracks could transit from the coating to the surface layer of the metal. In the gross slip mode (D = 30 µm and F = 3 N), the processes of abrasive interaction begin to prevail (Figure 12d). The shear relief of the coating along the bodies' direction motion is associated with the presence of a Pb soft component (Figure 12e). In this case, the lead plastically deformed during the slip, which reduced friction and wear. Figure 10f shows regions of the coating structure of differently oriented crystallites and "shear" regions, apparently corresponding to zones with a Pb high content.

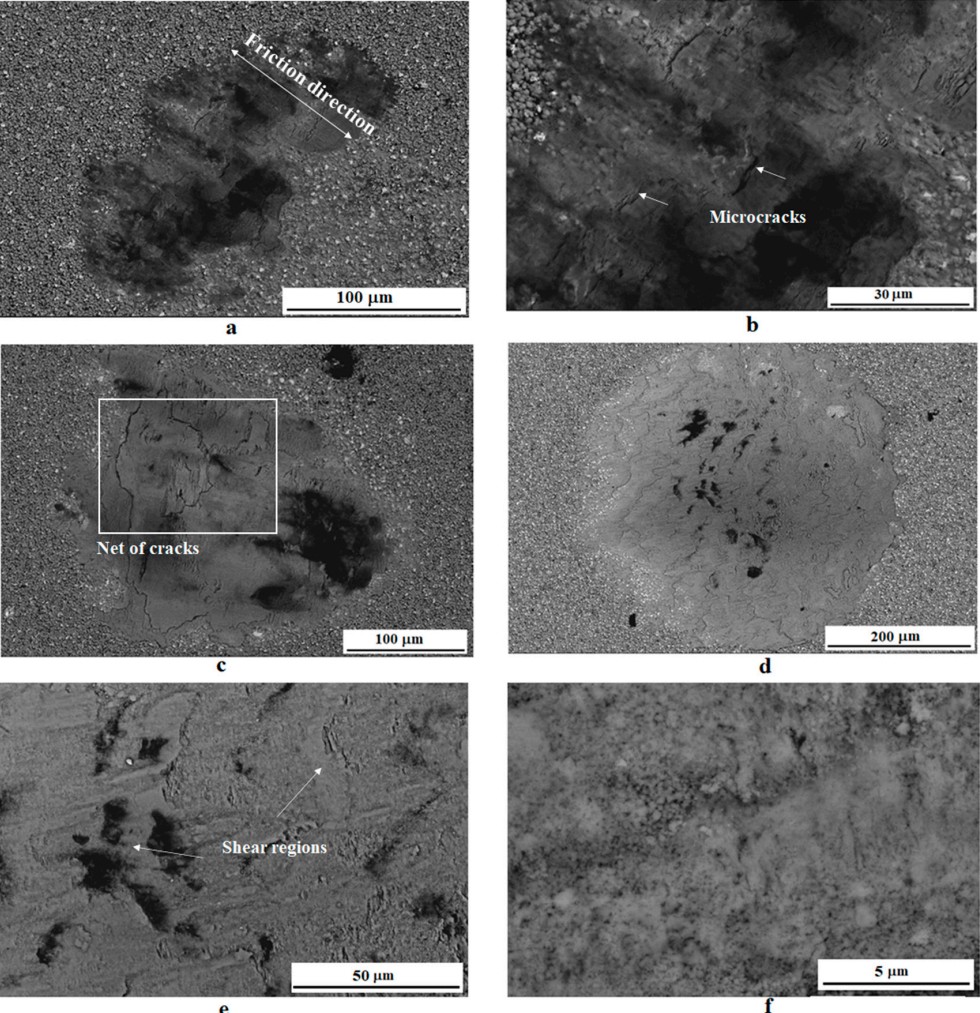

**Figure 12.** SEM images in backscattered electrons of TiN–Pb coating damage zone obtained at D = 5 µm and F = 13 N (**a,b**), at D = 15 µm and F = 10 N (**c**), at D = 30 µm and F = 3 N (**d–f**).

Elemental microanalysis of the wear patch obtained at loading parameters D = 15 μm, F = 10 N showed the transfer of the soft coating component to the edge region. It led to the coating preservation in this area and its destruction in the central regions accompanied by the formation of microcracks and the appearance of areas with increased titanium content (Figure 13a). The dissipation energy in the gross slip mode is the highest, which indirectly indicates a rather high wear rate. Elemental microanalysis of the wear patch shows the appearance of titanium zones up to 20 μm in size in the center (Figure 13) associated with the beginning of the process of complete coating wear to the metal. Mass transfer of the soft component is not observed in this mode.

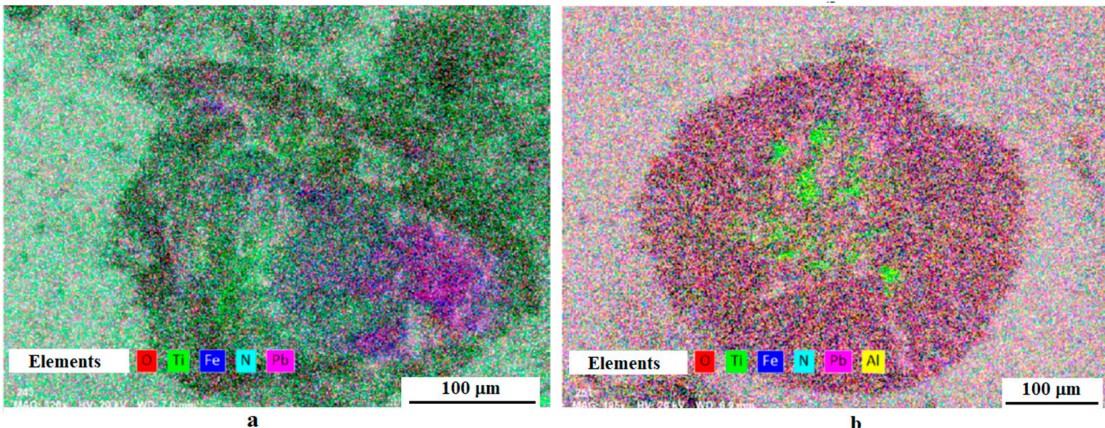

**Figure 13.** Summary maps of the elements' distribution of wear patches corresponding to loading parameters: D = 15 μm, F = 10 N (**a**) and D = 30 μm, F = 3 N (**b**).

The dissipation energy drops by more than 3 times, accompanied with a sharp decrease in the friction coefficient from 0.25 to 0.05 during the transition from the gross slip mode to the reciprocating slip mode. Thus, the reciprocating slip mode, as well as the partial slip mode, occur in narrow ranges of loads and displacements for TiN–Pb coating. The main mode of friction during the fretting is the gross slip mode. Mixed slip mode also occurs in a narrow range of the loads accompanied with the formation of a cracks network that can lead to its delamination. In the gross slip mode, the friction coefficient is set equal to ~0.25, which is significantly lower than the friction coefficient of TiN without Pb additives (~0.75–0.9) [25,26].

## 4. Discussion

The reactive deposition of TiN–Pb coatings is a multifactorial process complicated by a significant difference in the masses of Ti and Pb. The result is the different efficiency of the sputtered particles scattering process along the way to the substrate. Energy transfer during collisions of particles is maximum when their masses are equal. The masses of argon and titanium atoms are almost equal. As a result of successive collisions, Ti is effectively scattered by Ar slowing down and deviating at different angles. This effect ensures the deposition of titanium on the substrate by falling at an angle. On the other hand, nearly ballistic sputtering is realized for heavy lead atoms. They reach the substrates slightly deviating and losing a small amount of energy. By changing the angle of the ion beam, one can control such important properties of the coatings as structure, size of structural components, hardness, texture and morphology [47–49]. The structure of the formed coatings is usually also influenced by the substrate temperature, the chamber pressure, the initial surface roughness, and (especially at low temperature) the deposition rate and the angle of incidence of the deposited particles.

The deposition of sputtered Ti and Pb atoms on rotating substrates (coatings 1 and 2) provides their incidence angles in the range from 0 to 90° and leads to an increase in the coating thickness. This is due to such atoms involved in the process with direction of

arrival provided the self-shadowing effect. During the rotation and repeated passing of the substrate in front of the magnetrons, the atoms sputtered from the cathodes are deposited in a wide range of incidence angles from the normal (0–90°). This process is similar to the so-called "oscillation" deposition when the substrate oscillates in front of the sputtered target [50]. In this case, the substrate rotation rate is very important since the possibility of nucleation and growth direction of columnar crystallites also depends on it. Since the direction of vapor flow changes during the rotation, the direction of growth of individual columnar crystallites can change.

Sputtering of Pb in the medium frequency mode at 25 and 40 kHz for coatings 3 and 4 could contribute to their higher content in the coatings. The conditions of a stationary substrate and the medium-frequency Pb sputtering regime may promote to Pb-containing phases nuclei on the surface of TiN crystallites, their growth interruption and nanosized grains preservation. In addition, particles of Pb-containing phases could stimulate the formation of TiN nuclei of random orientations prevented the formation of a texture. A possible mechanism for this atomic densification is bombardment of the coating by high-energy argon atoms reflected from the cathode and sputtered heavy Pb atoms. This is due to the fact that both the average energy and the sputtered particles flux increase with the atomic mass of the target material.

Usually, TiN deposited coating has a pronounced columnar structure mostly combined with a pronounced crystallographic texture (111). TiN–Pb is a multicomponent coating. It has a structure formation mechanism different from two-component coatings. This is confirmed by changes in the crystallographic texture while in two-component coatings the texture is stably reproduced over a wide range of deposition coatings parameters. For example, it was found in [7,12] that the addition of Al and Si to TiN coating composition is accompanied by a transition of a pronounced texture (111) to a textureless state, microstructure refinement and the formation of a pseudo amorphous structure. As was found in [38], compressive stresses in the surface layer contribute to the formation of the deposited coating globular surface. As the lead content increases, the coating can become more fusible due to the presence of Pb and PbO phase inclusions. This can reduce the activation energy of diffusion processes. Under these conditions, volume diffusion becomes more significant, which can reduce internal stresses and prevent their accumulation. Increasing the fusibility of the coating can lead to an increase in the effects of recrystallization on the coating structure decreasing in the texture component as was obtained for coatings 2–4. An increase in the lead content with its uniform distribution in the coating can reduce the difference in thermal expansion of the substrate—steel and the coating with a TiN matrix. This can help to reduce the thermal stresses and the height of irregularities—the "globularity" of the surface. In this work it was found a decrease in the roughness of the coatings from 1 to 3. The composite structure of coating 4 did not lead to a decrease in roughness. It may indicate that a higher content of lead and Pb-containing phases in the TiN matrix can also lead to an increase in thermal stresses, phase migration of a soft and low-melting component into the surface layer [38].

Among the obtained coating (columnar, columnar nanostructured, textureless, nanostructured composite), nanostructured composite coating 4 showed the best tribological characteristics. In coating 4, the soft component is distributed both in the matrix and in the form of island-like inclusions predominantly in the surface layer. In columnar coatings (1 and 2), the crystallites are bound by a rather small area to the surface. In this case, the application of a load in the tangential direction can lead to their destruction. In addition, in the regions between the crystallites, there are often vertical pores in the form of channels through which the penetration of external environment products is possible. Therefore, the formation of a nanostructured textureless coating contributes to longer durability. Certainly, the friction coefficient, microhardness and roughness were essential in the testing process. In this case, the higher tribological characteristics stability of the composite nanostructured coating can be explained by the presence of soft island-like inclusions in the surface layer. This structure provides easier formation of shear regions under the loads, and the lower

content of lead in the matrix can maintain a relatively high microhardness. In the same time, a high content of Pb in the matrix (coating 3) can negatively influence wear resistance due to a decrease in microhardness, despite the low friction coefficient.

## 5. Conclusions

Solid lubricating ceramic composite TiN coatings with Pb additives were obtained on stationary and rotating substrates made of AISI 304 steel and titanium in the process of reactive magnetron sputtering of separate cathodes.

Columnar, columnar nanostructured, textureless and composite nanostructured TiN–Pb coatings with different Pb content (3–13%) were obtained by varying the deposition regimes. The columnar structure changes to a columnar nanostructured with an increase in the Pb content in the coating to 8%.

Deposition on a stationary substrate and the use of medium frequency sputtering of Pb lead to an increase in its content in the coatings and textureless state. The coatings include Pb, PbO and TiN phases. The diffraction lines are characterized by significant broadening, indicating nanometer size of the subgrains.

Tribological tests of the coatings were carried out at room temperature and under conditions of stepwise heating, as well as at various loading parameters. Among the obtained coating structures, the nanostructured composite coating showed the best tribological characteristics. In this coating the lead-containing components are distributed both in the matrix and in the form of inclusions contributed to relatively high microhardness (817 HV), a high Pb content and a textureless state with a low grain size. This provided a low friction coefficient (~0.1) over 50,000 test cycles, both at room temperature and under conditions of stepwise heating up to 100 °C and 200 °C.

**Author Contributions:** Conceptualization, A.L. and S.S.; methodology, I.N. and S.B.; software, E.K.; validation, M.L.; formal analysis, A.L.; investigation, S.S. and S.B.; resources, M.L.; data curation, A.L.; writing—original draft preparation, S.S. and A.L.; writing—review and editing, M.L.; visualization, E.K. and I.N.; supervision, A.L.; project administration, S.S.; funding acquisition, S.S. All authors have read and agreed to the published version of the manuscript.

**Funding:** This research was funded by Russian Science Foundation, project number 22-19-00754.

**Institutional Review Board Statement:** Not applicable.

**Informed Consent Statement:** Not applicable.

**Data Availability Statement:** Not applicable.

**Conflicts of Interest:** The authors declare no conflict of interest.

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
