# Peer review of "Investigation of Structural and Tribological Characteristics of TiN Composite Ceramic Coatings with Pb Additives"

_coatings, doi:10.3390/coatings13081463_

Round 1
Reviewer 1 Report
OVERALL RATING
The submitted manuscript presents a study concerning the tribological behavior at elevated temperatures of TiN coatings added with Pb deposited by reactive magnetron co-sputtering on 304 austenitic stainless steel and titanium substrates. The authors have evaluated the morphology, structure, thickness, roughness, and microhardness of the resultant coatings. They have conducted reciprocating wear tests at room temperature, 100 oC, and 200 oC. I think it is a valuable paper. However, a couple of points should be addressed to improve its quality.
SPECIFIC ANNOTATIONS
Materials and Methods
1. Page 4, lines 1147-148 >>> Please give details on roughness measurements. Did the authors conduct 2-D profile measurements or 3-D measurements to assess the coating surface topographies?
2. Page 4, line 152 >>> Please refer to “Ball-on-disk” instead of “Sphere-disk” tribological tests.
3. Page 4, line 154 >>> Please give the total sliding distance of the reciprocating sliding tests.
4. Page 4, line 155 >>> Please specify the type (grade designation) of steel used as counterbody, according to AISI, SAE, or ISO standards.
5. Page 4, lines 161-162 >>> What do the authors mean by “the mechanism of bodies interaction were analyzed in the testing process.”? Do the authors analyze the wear mechanisms?
6. Page 4 >>> Please give details of how the wear volume (“volumetric wear”) of the coatings and the material transfer from the counterbody were measured. See Fig. 7 page 10.
3.2 Surface Roughness
7. Page 6 >>> To distinguish between 2-D and 3-D parameters, use the headed letter ‘S’ (for ‘surface’) in the case of 3-D measurement or instead the letter ‘R’ for 2-D measurement, according to your answer to the first question.
3.6. Tribological Tests at Room Temperature
8. Page 10, Figs. 6 and 8 >>> Please plot friction coefficient as a function of the sliding distance and analyze these results along the text considering the sliding distance.
9. Page 10, lines 298-304 >>> Please verify the paragraph format.
3.7. Tribological Tests under Stepwise Heating Conditions
10. Page 11, line 310 >>> Use 2 h instead of “2 hours”.
11. Page 13, Fig. 11 >>> Axis ‘F, H’ ?
Author Response
The authors kindly thank to Reviewer for a detailed study of the work and the comments.
1. Page 4, lines 1147-148 >>> Please give details on roughness measurements. Did the authors conduct 2-D profile measurements or 3-D measurements to assess the coating surface topographies?
2D profiles were used to obtain the roughness parameters. 3D images were used only for visual comparison of coatings surfaces.
2. Page 4, line 152 >>> Please refer to “Ball-on-disk” instead of “Sphere-disk” tribological tests.
It was corrected in text.
3. Page 4, line 154 >>> Please give the total sliding distance of the reciprocating sliding tests.
The total sliding distance was 1.5 m with a displacement of 15 µm. It was added in the text.
4. Page 4, line 155 >>> Please specify the type (grade designation) of steel used as counterbody, according to AISI, SAE, or ISO standards.
Steel type (100Cr6) was added.
5. Page 4, lines 161-162 >>> What do the authors mean by “the mechanism of bodies interaction were analyzed in the testing process.”? Do the authors analyze the wear mechanisms?
An analysis of the interaction mechanism of friction bodies by the form of hysteresis loops by varying amplitude and load was carried out (Figure 11). Wear analysis was also carried out according to friction surfaces condition (Figure 12).
6. Page 4 >>> Please give details of how the wear volume (“volumetric wear”) of the coatings and the material transfer from the counterbody were measured. See Fig. 7 page 10.
Volumetric wear was analyzed using a confocal laser microscope LEXT OLS5000. The microscope software evaluates volumetric wear by integrating wear worn space points relative to the original surface. The transferred material of the counter body was estimated as the build-up volume in the wear patches. The build-up height exceeded the original level of the coating surface. In the case of its formation the build-up volume corresponded to the volumetric wear of the counter body.
This description was added in text.
7. Page 6 >>> To distinguish between 2-D and 3-D parameters, use the headed letter ‘S’ (for ‘surface’) in the case of 3-D measurement or instead the letter ‘R’ for 2-D measurement, according to your answer to the first question.
2D profiles were used to obtain the roughness parameters. 3D images were used only for visual comparison of coatings surfaces.
8. Page 10, Figs. 6 and 8 >>> Please plot friction coefficient as a function of the sliding distance and analyze these results along the text considering the sliding distance.
The authors agree with the reviewer that it would be really clearer if different regimes with a change in the loading parameters compared. However, in these graphs, the experiment results were compared in the same loading parameters regime. The total sliding distance (1.5 m) was added in the text.
9. Page 10, lines 298-304 >>> Please verify the paragraph format.
The paragraph format was corrected.
10. Page 11, line 310 >>> Use 2 h instead of “2 hours”.
It was corrected in text.
11. Page 13, Fig. 11 >>> Axis ‘F, H’ ?
The authors thank to Reviewer for the found mistake in this figure. H was corrected to N (Newtons).
Reviewer 2 Report
The authors present a paper dealing with the solid lubricating composite TiN coatings with Pb additives using a reactive magnetron sputtering method. They used different Pb percentages (3-13%) deposited on rotating and stationary steel and Ti substrates. They obtained the coatings with different mechanical features and tribological properties. They showed that the coating with high Pb content presents the best tribological feature.
The authors provide a well-structured and designed paper. The topic is also interesting and relevant for the reader. However, some comments should be addressed to improve the quality of this manuscript:
1) Can the authors explain why they used Pb as an additive element, providing that Pb is considered heavy metal and toxic?
2) Can the authors explain why they used different scales for showing SEM images? How comparison could be achieved between coatings?
3) In Table 2, is there any explanation for why the roughness increased between 8% and 12% of Pb and decreased for 13% of Pb?
4) Figure 4b, showing a color bar of the percentage of the different elements N, O Ti and Pb is recommended.
5) Can the authors explain why the thickness of the prepared coatings are not similar? May the thickness has also effect on the coatings properties?
e.g. The thickness of coating 1 = 5.8 µm and coatings 3 and 4 almost 2 µm.
6) Figure 13 a and b, can the author provide color bars to show the percentage of each element O, Ti, Fe, N and Pb, Al
Minor editing of English language is recommended
Author Response
The authors kindly thank the Reviewer for a detailed study of the work and the comments.
1)Can the authors explain why they used Pb as an additive element, providing that Pb is considered heavy metal and toxic?
Pb is the main component of solid lubricant coatings due to its unique tribological characteristics. TiN is an additional component that is added to increase its wear resistance. Despite the high specific weight and toxicity, lead is widely used in mechanical engineering in cases where it cannot be effectively replaced by other materials, in particular in high-machinability steels. In addition, it can be used, for example, in industries with minimal environmental influence, such as the space industry.
2) Can the authors explain why they used different scales for showing SEM images? How comparison could be achieved between coatings?
The magnifications were chosen primarily for the possibility of the coatings structure comparation. Size scales differ due to the different thicknesses of the coatings.
3) In Table 2, is there any explanation for why the roughness increased between 8% and 12% of Pb and decreased for 13% of Pb?
The coating 4 differs in the distribution of lead. Lead is present both in the form of inclusions mainly in the surface layer of the coating 4 and in its matrix. Therefore, it may be less in the matrix than for the coatings without island inclusions with 8 and 13% lead. This may lead to a higher level of thermal stresses at the final formation stage of coating 4 and an increase in roughness.
4) Figure 4b, showing a color bar of the percentage of the different elements N, O Ti and Pb is recommended.
This elements distribution map does not contain quantitative information about the elemental composition, which was determined by the capabilities of the device.
5) Can the authors explain why the thickness of the prepared coatings are not similar? May the thickness has also effect on the coatings properties? e.g. The thickness of coating 1 = 5.8 µm and coatings 3 and 4 almost 2 µm.
The thickness of the coatings certainly affected the properties of the coatings and differed with varying the deposition parameters. It was important to show that coating 4 with a thickness of almost 2 times less demonstrated better tribological characteristics than the more traditional columnar coating 1.
6) Figure 13 a and b, can the author provide color bars to show the percentage of each element O, Ti, Fe, N and Pb, Al
This elements distribution map does not contain quantitative information about the elemental composition, which was determined by the capabilities of the device.
Reviewer 3 Report
In this work, the authors synthesized TiN coatings with different microstructure by varying concentrations of Pb additives and adjusting the deposition parameters and then characterized their microstructure by XRD and SEM, as well as tribological tests. This work could provide a reference for future solid lubricant coating design. After reviewing, I would suggest accepting it after the following minor comments are addressed.
1. Line 10-13, it would be better to revise the statement by “Columnar, columnar nanostructured, and composite nanostructured TiN coatings with different contents (3–13%) Pb of a lubricating component have been obtained by deposition onto rotating and stationary substrates.”
2. Line 20, 252-“10÷20 nm”, Line 166, etc., please use the correct symbol here and in the manuscript.
3. Line 135-136, why did the authors design a different total deposition time in coating 1? It would lead to completely different residual stress and further affect the hardness, and tribological behavior. In other words, different deposited parameters (such as Ar flow rate….) and different Pb content can lead to the distinct microstructure of the coating. Which factors dominate the coating microstructure in this work? Please clarify these two questions in the manuscript.
4. Line 149-151, 177, why did the authors design coatings 1-4 with different coating thicknesses (Table 3)? Since the authors applied the same load in the hardness test, was the substrate effect taken into account during indentation? Please demonstrate in the manuscript.
5.Line 153 and 160, why did the authors design such a short stroke length (micrometer level, while the counter body is millimeter level) during the tribological test?
6. Line 192-195, the low surface roughness of Coating 3 is ascribed to the low thermal stress. Why does coating 3 have a lower thermal stress, considering that coating 2-4 have the same deposition time and similar elements?
7. Line 210, the Ra value of coating 4 is the highest, compared to the other three coating samples. Please double-check and revise this statement.
8. Line 229-230, please revise the statement “as in coating 1 (Figure 4c).” to as in coating 1 (Figure 4a).
9. Figure 4 and Figure 9, please add the scale bars in the inserted images of Figure 4(a) and (e), and Figure 9(a) and (b).
10. In Table 3, the hardness reduction of coating 3 is interesting, the authors’ hypothesis is “An increase in Pb content can lead to the formation in the coating structure of such compounds as lead oxides, lead oxynitride, Ti3PbO7, TiPbO3, etc. [42]. The hardness of these compounds is significantly inferior to TiN, which contributes to a general decrease in hardness”. However, the concentration of Pb in coating 4 is 12 %, just slightly lower than in coating 3 (13%). Why do these compounds have no detrimental effect on the hardness of Coating 4?
11. Please revise the font and size in Lines 298-304.
12. Line 361-the title of Figure 11, “Fretting map of nanostructured composite coating 4 on titanium.”, why is the substrate titanium here instead of steel?
Author Response
The authors kindly thank the Reviewer for a detailed study of the work and the comments.
1. Line 10-13, it would be better to revise the statement by “Columnar, columnar nanostructured, and composite nanostructured TiN coatings with different contents (3–13%) Pb of a lubricating component have been obtained by deposition onto rotating and stationary substrates.”
It was corrected in text.
2.Line 20, 252-“10÷20 nm”, Line 166, etc., please use the correct symbol here and in the manuscript.
The symbols were corrected in text.
3.Line 135-136, why did the authors design a different total deposition time in coating 1? It would lead to completely different residual stress and further affect the hardness, and tribological behavior. In other words, different deposited parameters (such as Ar flow rate….) and different Pb content can lead to the distinct microstructure of the coating. Which factors dominate the coating microstructure in this work? Please clarify these two questions in the manuscript.
Authors tried to obtain TiN-Pb coatings that differ in their structure and properties based on our experience in obtaining TiN coatings and from other components. The aim of the work was to search for correlations between the structure and tribological properties of coatings, which could be realized varying the deposition conditions and coating composition. The obtained regularities make it possible to carry out a more purposeful choice of the conditions for deposition of coatings and their composition.
4.Line 149-151, 177, why did the authors design coatings 1-4 with different coating thicknesses (Table 3)? Since the authors applied the same load in the hardness test, was the substrate effect taken into account during indentation? Please demonstrate in the manuscript.
The authors thank to Reviewer for the important remark. Really, the influence of the substrate is a very important factor on the results of thin coatings hardness measuring. Therefore, we chose loads that ensure the penetration depth of the indenter much less than the thickness of the coatings (no more than 1/3 of the thickness).
5.Line 153 and 160, why did the authors design such a short stroke length (micrometer level, while the counter body is millimeter level) during the tribological test?
Authors analyzed the performance of a solid lubricant coating under fretting conditions, since their efficiency in low-amplitude friction conditions (fretting) is the most important applied problem.
6. Line 192-195, the low surface roughness of Coating 3 is ascribed to the low thermal stress. Why does coating 3 have a lower thermal stress, considering that coating 2-4 have the same deposition time and similar elements?
The coating 3 contains more soft component in the matrix (13%) compared to coating 4, in which lead is present also in the form of islands, and coating 2, where there is less lead in the matrix (8%).
7. Line 210, the Ra value of coating 4 is the highest, compared to the other three coating samples. Please double-check and revise this statement.
The roughness of coating 4 is really higher. The coating 4 differs in the distribution of lead. Lead is present both in it in the form of inclusions mainly in the surface layer of the coating and in matrix. Therefore, it may be less in the matrix than for the coatings without island inclusions with 8 and 13% lead. This may lead to a higher level of thermal stresses at the final formation stage of coating 4 and an increase in roughness.
8. Line 229-230, please revise the statement “as in coating 1 (Figure 4c).” to as in coating 1 (Figure 4a).
The authors thank to Reviewer for the found mistake. It was corrected in text.
9. Figure 4 and Figure 9, please add the scale bars in the inserted images of Figure 4(a) and (e), and Figure 9(a) and (b).
The scale bars were added in Figures 4 and 9.
10. In Table 3, the hardness reduction of coating 3 is interesting, the authors’ hypothesis is “An increase in Pb content can lead to the formation in the coating structure of such compounds as lead oxides, lead oxynitride, Ti3PbO7, TiPbO3, etc. [42]. The hardness of these compounds is significantly inferior to TiN, which contributes to a general decrease in hardness”. However, the concentration of Pb in coating 4 is 12 %, just slightly lower than in coating 3 (13%). Why do these compounds have no detrimental effect on the hardness of Coating 4?
The coating 4 differs in lead distribution from other coatings. The average content of lead is 12% in the coating. Lead is present both in the matrix and in the form of inclusions mainly in the surface layer of the coating. At the same time, it is less in the matrix than for coatings with 8 and 13% lead, which leads to greater hardness.
11. Please revise the font and size in Lines 298-304.
The paragraph format was corrected.
12. Line 361-the title of Figure 11, “Fretting map of nanostructured composite coating 4 on titanium.”, why is the substrate titanium here instead of steel?
Coating 4 showed better tribological performance compared other coatings. It was also deposited and tested on 99.0%Ti substrate, as noted in Paragraph 2. Substrates Information was added in Table 1.
Reviewer 4 Report
The authors carried out interesting research on the microstructure and tribological test on the TiN composite that was prepared using the magnetron sputtering methods. The Pb-containing precipitates are shown to increase the microhardness and improve the tribological property. The systematic research in this work deserves publication in the journal Coatings. However, there are a few questions and comments requiring the authors to address.
1. In lines 117-118 on page 3, it states that ''Solid lubricant nanostructured TiN coatings with Pb additives were formed on steel 117 AISI 304 and titanium VT1 (99.0%Ti) samples...''. This is confusing. Did the authors deposit on the VT1 substrate as well? The authors should clarify this information by rewording this sentence and the others in the paper. It would also be better to list the substrate and targets information in Table 1 as well.
2. In lines 156-157 on page 4, it states that ''The influence of heating on the tribological properties of the coatings was studied in the 156 stepwise heating mode up to 100 and 200 °C with holding at each temperature for 2 h.'' Does it mean the coatings were annealed at 100 and 200 °C for 2 hours? what is the test temperature for the stepwise-heated sample?
3. The authors mentioned several times the coating layers have a growth texture in this paper. For example, in line 249 on page 8, a pronounced (111) crystallographic texture. It is interesting to see the growth texture in coatings 1 and 2. Can the authors roughly show more information about this growth texture? For example, what is the growth direction of the column in coatings 1 and 2?
4. In line 252 on page 8, it is shown that the size of the subgrains are in range ~10-20 nm. 10-20 nm nanostructure is extremely small. This information could be obtained from the XRD pattern fitting. However, it should be careful to claim this information. The size of the subgrains should be evaluated by using TEM for the current coating 4.
5. In lines 256-258 on page 8, it is said that ''In addition, particles of Pb and PbO phases stimulate the formation of TiN nuclei of random orientations, which prevents the formation of a pronounced texture.'' Is there any evidence to support this information or just a hypothesis in the current paper?
Author Response
The authors kindly thank the Reviewer for a detailed study of the work and the comments.
1.In lines 117-118 on page 3, it states that ''Solid lubricant nanostructured TiN coatings with Pb additives were formed on steel 117 AISI 304 and titanium VT1 (99.0%Ti) samples...''. This is confusing. Did the authors deposit on the VT1 substrate as well? The authors should clarify this information by rewording this sentence and the others in the paper. It would also be better to list the substrate and targets information in Table 1 as well.
Coating 4 showed better tribological performance compared other coatings. It was also deposited and tested on 99.0%Ti substrate, as noted in Section 2. Substrates Information was added in Table 1.
2. In lines 156-157 on page 4, it states that ''The influence of heating on the tribological properties of the coatings was studied in the 156 stepwise heating mode up to 100 and 200 °C with holding at each temperature for 2 h.'' Does it mean the coatings were annealed at 100 and 200 °C for 2 hours? what is the test temperature for the stepwise-heated sample?
The tribological tests were carried out after stepwise heating with exposure for 2 h at each temperature (100 and 200 °C). The test temperature was 25 °C.
3.The authors mentioned several times the coating layers have a growth texture in this paper. For example, in line 249 on page 8, a pronounced (111) crystallographic texture. It is interesting to see the growth texture in coatings 1 and 2. Can the authors roughly show more information about this growth texture? For example, what is the growth direction of the column in coatings 1 and 2?
In Paragraph 3.4 the correlation between morphology and crystallographic orientation of coatings is discussed. Earlier in our work [41], we showed an X-ray diffraction pattern of TiN-Pb coating with a columnar structure and clearly visible (111) texture. No other TiN reflections were observed apart from the (111) reflection. Figure 5 shows the X-ray pattern of non-columnar structure coating 4 (Fig. 4e.) This X-ray pattern contains almost all reflections corresponding to a given range of diffraction angles (111), (200), (220) with close intensities indicated the textureless state of the coating.4.
4. In line 252 on page 8, it is shown that the size of the subgrains are in range ~10-20 nm. 10-20 nm nanostructure is extremely small. This information could be obtained from the XRD pattern fitting. However, it should be careful to claim this information. The size of the subgrains should be evaluated by using TEM for the current coating 4.
Undoubtedly, TEM provides direct information on the size of subgrains. The broadening of X-ray diffraction lines provides information on the size of structural elements, which somewhat differ from the size of subgrains measured by TEM. However, numerous experimental comparisons of the X-ray and TEM measurements have shown that they are close, and therefore the X-ray method is widely used to estimate the sizes of coating structure elements.
5.In lines 256-258 on page 8, it is said that ''In addition, particles of Pb and PbO phases stimulate the formation of TiN nuclei of random orientations, which prevents the formation of a pronounced texture.'' Is there any evidence to support this information or just a hypothesis in the current paper?
Of course, this is a hypothesis, since similar studies are few in the literature. it is known that TiN coatings have a pronounced (111) texture in most cases. A similar effect was found in the works [7, 12]. The addition of metals (for example, Al, Zr) to the coating composition significantly weakened the texture, and the addition of Si or Cu even leaded to amorphization of the structure. These data formed the basis for this hypothesis, which certainly requires further research to confirm it.
Reviewer 5 Report
The authors of this article investigate the effect of the content of Pb additives on the structural and tribological characteristics of TiN composite ceramic coatings. There are still some problems in the article, and the author is recommended to improve it. It is suggested that this manuscript can be accepted after minor revisions. Listed below are detailed comments on the author's manuscript.
1. It is proposed that the "Introduction section" be elevated.
2. Page 1 – line 20, "…in range of 10÷20 nm" Please check the full text for any errors here.
3. What does the "●" in Figure 5 stand for?
4. Page 10 – line 298, "Figure 7. shows a …is much higher. " Is this a comment on the Figure 7 here? Please check.
5. It is recommended that the rulers be added to Figure 9.
6. The text in Figure 12a is too small to be recognized.
Author Response
The authors kindly thank the Reviewer for a detailed study of the work and the comments.
1. It is proposed that the "Introduction section" be elevated.
The authors have tried to include in Section 1 the main works devoted to the structure and properties of similar coatings with oxide and nitride matrix with the addition of soft metal components. Unfortunately, at present these works are not numerous.
2. Page 1 – line 20, "…in range of 10÷20 nm" Please check the full text for any errors here.
The symbols were corrected in text.
3. What does the "●" in Figure 5 stand for?
The marks refer to the reflections from the AISI 30 steel substrate with an FCC crystal lattice.
4. Page 10 – line 298, "Figure 7. shows a …is much higher. " Is this a comment on the Figure 7 here? Please check.
The authors thank to Reviewer for the found mistake. The paragraph format was corrected.
5. It is recommended that the rulers be added to Figure 9.
The scale bars were added in Figure 9.
6. The text in Figure 12a is too small to be recognized.
The font size has been increased in Figure 12a.